# LncRNA MIR4435-2HG mediates cisplatin resistance in HCT116 cells by regulating Nrf2 and HO-1

Ping Luo[1☯], Shugui Wu[2☯], Kaibao Ji[3☯], Xia Yuan[4], Hongmi Li[4], Jinping Chen[5], Yunfei Tian[6], Yang Qiu[4]*, Xiaoming Zhong[4]*

1 Department of Breast Tumor of Nanchang Third Hospital, Nanchang, Jiangxi, China, 2 Department of Oncology of Ganzhou People's Hospital, Ganzhou, Jiangxi, China, 3 Department of Ophthalmology, Renmin Hospital of Wuhan University, Hubei, China, 4 Department of Tumor Radiotherapy of Jiangxi Province Cancer Hospital, Nanchang, Jiangxi, China, 5 Department of Oncology of Yichun People's Hospital, Yichun, Jiangxi, China, 6 Department of Interventional of Ganzhou People's Hospital, Ganzhou, Jiangxi, China

☯ These authors contributed equally to this work.
* QY416523616629@163.com (YQ); jddxf2012@126.com (XZ)

**Data Availability Statement:** The analyzed datasets generated during the study are available at the following doi:10.5061/dryad.djh9w0vxc.

## Abstract

### Purpose

Cisplatin resistance is still a serious problem in the clinic. However, the underlying mechanism remains unknown. In our study, we investigated cisplatin resistance by using the cisplatin-resistant cell line HCT116R.

### Methods

The HCT116 cell line, a colon cancer cell line, was purchased. Cell viability was determined using CCK-8 Assay Kit. The gene expression levels of MIR4435-2HG, Nrf2, and HO-1, and caspase activity were determined using qRT-PCR and Caspase 3 Assay Kit, respectively.

### Results

In this study, we found that the levels of the lncRNA MIR4435-2HG were dramatically increased in the cisplatin-resistant cell line HCT116R. Knockdown of MIR4435-2HG in HCT116R cells significantly restored the sensitivity to cisplatin, inhibited cell proliferation and promoted cell apoptosis. Furthermore, Nrf2 and HO-1 mRNA levels, as critical molecules in the oxidative stress pathway, were inhibited by siRNAs targeting MIR4435-2HG, suggesting that MIR4435-2HG-mediated cisplatin resistance occurs through the Nrf2/HO-1 pathway.

### Conclusion

Our findings demonstrate that the lncRNA MIR4435-2HG is a main factor driving the cisplatin resistance of HCT116 cells.

**Funding:** This work was supported by the grants from the Natural Science Foundation of Jiangxi Province of China (No. 20161BAB205272).

**Competing interests:** The authors declare that they have no competing interests.

**Abbreviations:** CRC, colorectal cancer; HO-1, Heme oxygenase-1; lncRNAs, Long noncoding RNAs; Nrf2, Nuclear factor erythroid 2-related factor; qRT-PCR, Quantitative real-time polymerase chain reaction.

## Introduction

Colon cancer is one of the most common malignant tumors in the world. At present, there are more than 1 million new cases of colon cancer each year, which seriously endangers human life and the quality of life [1]. Most patients with metastatic colorectal cancer cannot be cured. The positive response ratio to combined chemotherapy for colon cancer is only 20–47%, and most patients have recurrence [2]. Cisplatin is widely used to treat a variety of cancers [3]. However, drug resistance frequently occurs after a period of administration and an effective response, and its specific mechanism is still not very clear [4]. It has been shown that long non-coding RNA (lncRNA) interacts with chromatin regulatory proteins, RNA-binding proteins, and small RNAs to form a functional complex, regulating multiple important biological processes [5–7]. To investigate the role of the lncRNA MIR4435-2HG in cisplatin resistance, we disrupted the expression of MIR4435-2HG in cisplatin-resistant HCT116R cells, and these cells became sensitive to cisplatin. At the same time, Nrf2 and HO-1 mRNA levels also decreased.

## Materials and methods

### Cell culture and reagents

The colon cancer cell line HCT116 was purchased from the American Type Culture Collection (Manassas, VA, USA). This is a cisplatin-sensitive cell line, and a cisplatin-resistant cell line, HCT116R, was established from the parental HCT116 cells by selection with cisplatin in the laboratory. HCT116 cells were seeded at a density of $2.5 \times 10^5$ cells /well and subjected to treatment with 25 μM cisplatin for three months. The culture medium containing 25 μM cisplatin was renewed twice a week. Both HCT116 and HCT116R cells were maintained as a monolayer in McCoy's 5A medium (Gibco, Life Technologies, USA) supplemented with 10% fetal bovine serum, penicillin (100 U/ml), and streptomycin (100 μg/ml) in a humidified atmosphere of 5% $CO_2$ at 37˚C. Scramble and MIR4435-2HG siRNAs were purchased from Thermo Fisher (China), and siRNA transfections were performed with Lipofectamine 2000 (Life Technologies).

### Cell proliferation assays

The cell proliferation procedure was performed as previously described [8]. Briefly, cell proliferation was determined with WST-8 assay using Cell Counting Kit-8 (CCK-8) (Dojindo Laboratories, Kumamoto, Japan). HCT116 cells ($5 \times 10^3$/well) were seeded in 96-well plates and incubated for 24 h. Cells were incubated with or without 25 μM cisplatin for 24 h in McCoy's 5A medium. Each treatment was carried out in triplicates. After adding 10 μL CCK-8 reagent to each well, the plates were incubated at 37˚C and 5% $CO_2$ for another 4 h. Then, absorbance was determined at 450 nm using Microplate Reader (Multiskan JX; MTX Lab Systems, Vienna, VA, USA).

### Total RNA isolation and quantitative real-time polymerase chain reaction (qRT-PCR)

The procedure for this section has been described previously [9]. Total RNA was isolated from treated cells using Trizol Reagent (Invitrogen, Carlsbad, CA) following manufacturer's guidelines. RNA concentration and quality were determined by spectrophotometer (Beckman, Brea, CA) and gel electrophoresis, respectively. First-strand cDNA was synthesized using the M-MLV Reverse Transcriptase kit (Invitrogen, Carlsbad, CA) in a total volume of 20 ul. qRT-PCR analysis for gene expression was performed in triplicate using SYBR Green.

## Caspase-3 activity assay

Cells were seeded into 96-well plates at a density of 5,000 cells/ well and allowed to grow for 24h. Then, siRNAs targeting MIR4435-2HG were transfected by Lipofectamine RNAiMAX (ThermoFisher, China) according to the instruction manual. Cells were harvested after 24 h, and caspase activity was examined by the Caspase-3 Assay Kit (Colorimetric) (Abcam ab39401, Cambridge, UK).

## Statistical analysis

Data were presented as the mean± standard error of the mean. Statistical analysis was performed using Prism GraphPad 6.0 (GraphPad Software, San Diego, CA, USA). $p <0.05$ was considered significant.

## Results

### The MIR4435 level is increased in HCT116 cells resistant to cisplatin

First, we induced the drug resistance of HCT116 to cisplatin by adding 25 μM cisplatin to HCT116 cells for 3 months, and we named the resulting line HCT116R. HCT116R cells proliferated in the presence of 25 μM cisplatin, while most HCT116 cells progressively died (Fig 1A). We detected the response to cisplatin. Then, We found that the level of the lncRNA MIR4435-2HG dramatically increased by approximately ten times, and this result was verified with real-time PCR (Fig 1B).

### Knockdown of MIR4435-2HG improved the sensitivity of HCT116R cells to cisplatin

To investigate the relationship between the lncRNA MIR4435-2HG and cisplatin resistance, we knocked down MIR4435-2HG by siRNA in HCT116R cells, and the MIR4435-2HG level dramatically decreased after siRNA transfection, which was verified by real-time PCR (Fig 2A). The knockdown of MIR4435-HG2 by siRNA increased the sensitivity of HCT116R cells to cisplatin, and the cell viability was tested by WST assay. In the presence of 25 μM cisplatin, HCT116R cell proliferation was inhibited by the siRNA targeting MIR4435-2HG, while there was no significant effect with the scrambled siRNA sequence (Fig 2B). After knockdown of

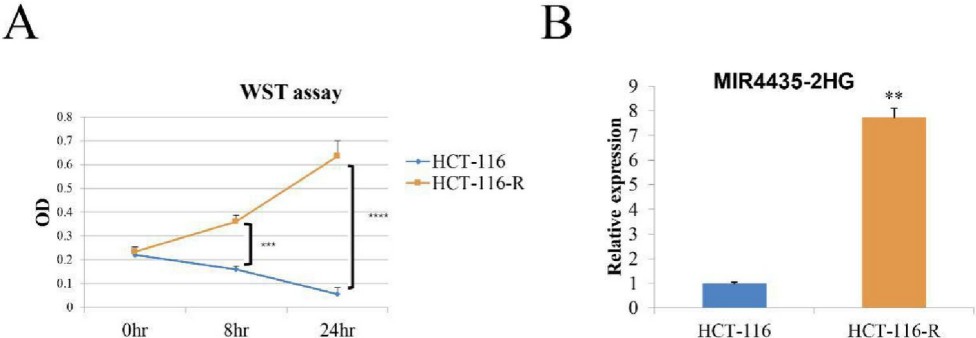

**Fig 1. The MIR4435 level was increased in the HCT116 cell line resistant to cisplatin.** A. HCT116 and HCT116R cells were treated with 25 μM cisplatin, and the cell viability was tested by the WST assay 0, 8 and 24 h later. B. MIR4435-2HG levels in HCT116 and HCT116R cells were detected by real-time PCR. ** P < 0.01, *** P < 0.001, **** P < 0.0001.

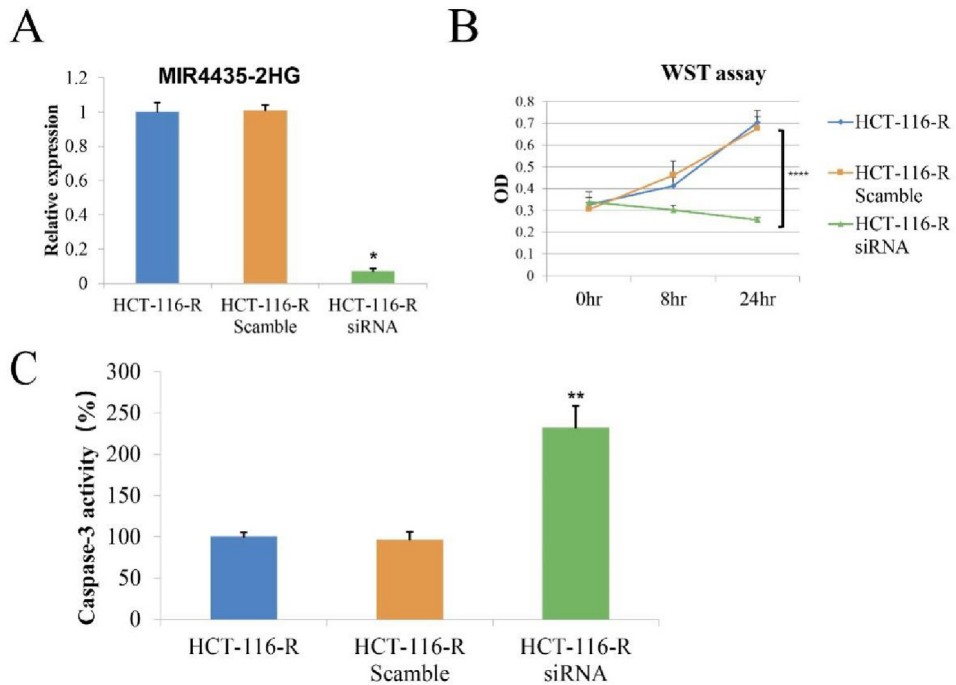

**Fig 2. Transient knockdown of MIR4435 by siRNA sensitized HCT116R cells to cisplatin.** A. MIR4435-2HG was knocked down in HCT116R cells by siRNA, and the effect of siRNA was verified by real-time PCR. B. Knockdown of MIR4435-HG2 by siRNA increased the sensitivity of HCT116R cells to cisplatin, and the cell viability was tested by the WST assay 0, 8 and 24 h after the administration of cisplatin. C The caspase-3 activity was tested 24 h after the cisplatin treatment. The data are presented as the means ± SDs. * P < 0.05, ** P < 0.01, **** P < 0.0001.

MIR4435-2HG, cisplatin administration increased caspase-3 activity approximately 2-fold (Fig 2C), which indicated that HCT116R cell apoptosis occurred.

## Knockdown of MIR4435-2HG decreased Nrf2 and HO-1 mRNA levels related to oxidative damage

Nuclear factor erythroid 2-related factor 2 (Nrf2) is a key transcription factor that regulates antioxidant and detoxification enzymes, and heme oxygenase-1 (HO-1) is an Nrf2-regulated gene which performs a crucial role in the precaution of inflammation. Transfection with MIR4435-2HG siRNA downregulated the mRNA levels of Nrf2 and HO1 significantly after treatment with 25 μM cisplatin (Fig 3), and this phenomenon indicates that MIR4435-2HG is involved in oxidative stress.

## Discussion

Cisplatin is thought to generate covalent adducts between some bases in DNA and can cure certain kinds of testicular and ovarian carcinomas [10, 11]. However, the efficiency of cisplatin is low in treatment of colorectal cancer (CRC), and less than 20% of patients achieved clinical response using it alone or in combination with other drugs [12]. Over the last few years, several studies have suggested that some lncRNAs contribute to cisplatin resistance via various mechanisms[13].

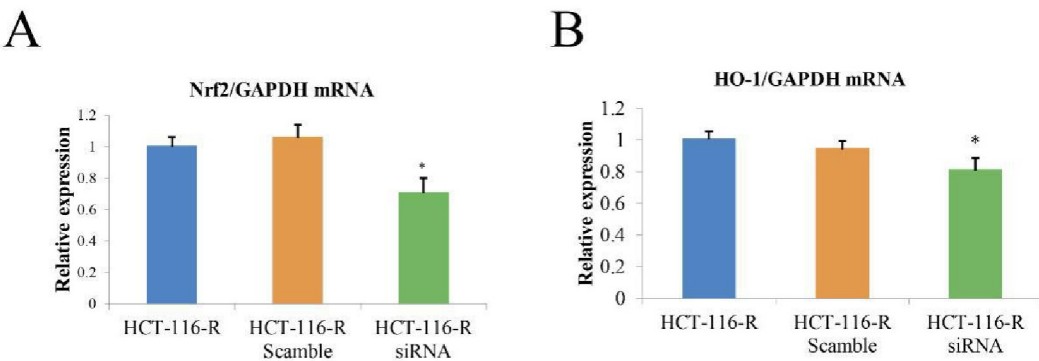

**Fig 3. Knockdown of MIR4435 decreased the NRF2 and HO-1 mRNA levels related to oxidative damage.** The relative NRF2 (A) and HO-1 mRNA (B) levels of HCT116R cells were analyzed by real-time PCR, and the results were normalized to the reference gene, GAPDH. The data are presented as the means ± SDs. * P < 0.05.

The lncRNA MIR4435-2HG has been reported to be related to hepatocellular carcinoma [14] and lung cancer [15]. Wen et al. found that MIR4435-2HG might participate in the development of colorectal cancer via the P38/MAPK and VEGF pathways[16]. In this report, we found that the level of MIR4435-2HG was increased approximately 7-8-fold in cisplatin-resistant cell lines compared to its expression in normal cell lines, which implies that MIR4435-2HG may be involved in cisplatin resistance. Then, we knocked down MIR4435-2HG with siRNA, and approximately 90% of MIR4435-2HG expression was inhibited by siRNA in cisplatin-resistant HCT116R cells. The effect of MIR4435-2HG knockdown in HCT116R cells was that 24 h of cisplatin administration caused approximately one-third of HCT116R cells to die. As a control, HCT116R cells without disruption of MIR4435-2HG expression proliferated approximately twice as much in 24 h. The cisplatin-resistant HCT116R cells became sensitive to cisplatin after MIR4435-2HG knockdown. In addition, after 24 h of cisplatin treatment, the caspase-3 activity increased 2.5-fold in the MIR4435-2HG knockdown cells compared to that in HCT116R cells, while the scramble siRNA did not have any effect. These results indicate that cisplatin-induced cell death may mainly occur via apoptosis.

Nuclear factor erythroid 2-related factor 2 (Nrf2), a transcription factor, responds to oxidative stresses and plays a key role in redox homeostasis [17]. Heme oxygenase (HO), acting downstream of Nrf2 [18], is an enzyme that catalyzes the degradation of heme [19, 20]. These two components are usually increased in different types of tumors and correlate with tumor progression, aggressiveness, resistance to therapy, and poor prognosis [21]. In our system, the knockdown of MIR4435-2HG significantly decreased NRF2 and HO-1 mRNA levels.

## Conclusions and perspectives

In this report, we observed that the lncRNA MIR4435-2HG was a main factor driving cisplatin resistance in HCT116 cells. Moreover, MIR4435-2HG might contribute to the development of cisplatin resistance through the Nrf2/HO-1 pathway. The current study discovered the roles of MIR4435-2HG in colorectal cancer cisplatin resistance and provided the underlying mechanism of cisplatin resistance.

By increasing our understanding of the underlying mechanisms of cisplatin resistance, we can develop better colorectal treatments with lower treatment resistance by adopting the following strategies. Firstly, new platinum-based drug research and development should be

expedited. Secondly, studies should find methods to increase the efficiency of cisplatin delivery to tumors. Thirdly, experimental research should be carried out to target cisplatin resistance mechanisms. Lastly, integrating cisplatin with other drugs should be considered. A greater understanding of cisplatin resistance mechanism can help identify which patients respond to therapy. Therefore, oncologists can make the best-informed decision when choosing effective therapy for patients.

## Acknowledgments

We thank Chaoming Zhou for helpful assistance.

## Author Contributions

**Conceptualization:** Kaibao Ji, Yang Qiu.

**Data curation:** Ping Luo, Shugui Wu, Kaibao Ji, Yang Qiu.

**Formal analysis:** Ping Luo, Shugui Wu, Kaibao Ji, Yang Qiu.

**Funding acquisition:** Yang Qiu, Xiaoming Zhong.

**Investigation:** Shugui Wu.

**Methodology:** Shugui Wu.

**Resources:** Shugui Wu, Yunfei Tian.

**Software:** Ping Luo, Jinping Chen.

**Validation:** Yunfei Tian.

**Visualization:** Shugui Wu, Jinping Chen, Yunfei Tian.

**Writing – original draft:** Xia Yuan, Hongmi Li.

**Writing – review & editing:** Xiaoming Zhong.

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
