## [Decision Letter · Decision Letter 0]

12 Dec 2019

PONE-D-19-25514

LncRNA MIR4435-2HG mediates cisplatin resistance in HCT116 cells by regulating Nrf2 and HO-1

PLOS ONE

Dear Mr. Zhong,

Thank you for submitting your manuscript to PLOS ONE. After careful consideration, we feel that it has merit but does not fully meet PLOS ONE’s publication criteria as it currently stands. Therefore, we invite you to submit a revised version of the manuscript that addresses the points raised during the review process.

Please consider the comments of Reviewer #1 before submitting a revised manuscript.

We would appreciate receiving your revised manuscript by Jan 26 2020 11:59PM. To enhance the reproducibility of your results, we recommend that if applicable you deposit your laboratory protocols in protocols.io, where a protocol can be assigned its own identifier (DOI) such that it can be cited independently in the future. For instructions see: http://journals.plos.org/plosone/s/submission-guidelines#loc-laboratory-protocols

We look forward to receiving your revised manuscript.

Kind regards,

Salvatore V Pizzo

Academic Editor

PLOS ONE

Journal Requirements:

1. We noticed you have some minor occurrence of overlapping text with the following previous publication(s), which needs to be addressed:

https://doi.org/10.1111/jgh.13069

https://doi.org/10.1097/PGP.0000000000000079

https://doi.org/10.7717/peerj.6683

https://doi.org/10.2147/DDDT.S106412

In your revision ensure you cite all your sources (including your own works), and quote or rephrase any duplicated text outside the Methods section. Further consideration is dependent on these concerns being addressed.

2. Please note that all PLOS journals ask authors to adhere to our policies for sharing of data and materials: https://journals.plos.org/plosone/s/data-availability. According to PLOS ONE’s Data Availability policy, we require that the minimal dataset underlying results reported in the submission must be made immediately and freely available at the time of publication. As such, please remove any instances of 'unpublished data' or 'data not shown' in your manuscript and replace these with either the relevant data (in the form of additional figures, tables or descriptive text, as appropriate), a citation to where the data can be found, or remove altogether any statements supported by data not presented in the manuscript. Particularly, either include as a Supporting file the RNA-seq data you mention at the beginning of your Results section and add the corresponding methodology in the Methods section, or consider removing the sentence altogether.

Reviewers' comments:

Reviewer's Responses to Questions

**Comments to the Author**

1. Is the manuscript technically sound, and do the data support the conclusions?

Reviewer #1: Yes

2. Has the statistical analysis been performed appropriately and rigorously? 

Reviewer #1: Yes

3. Have the authors made all data underlying the findings in their manuscript fully available?

Reviewer #1: Yes

4. Is the manuscript presented in an intelligible fashion and written in standard English?

Reviewer #1: Yes

5. Review Comments to the Author

Reviewer #1: The Cohen's d factors are large enough so the results are clear and unambiguous. The results may provide the useful tool how to circumvent the resistance to cisplatin beside the determination of polimorfisms of proteins involved in transport of cisplatin.

6. PLOS authors have the option to publish the peer review history of their article (what does this mean?). If published, this will include your full peer review and any attached files.

Reviewer #1: Yes: Tomaž Makovec

---

## [Author Response · Author response to Decision Letter 0]

3 Jul 2020

Dear Tomaž Makovec and Salvatore V Pizzo,

Thanks for your comments. We really appreciate your hard work. We really hope our manuscript can be published in your journal. Thank you for your consideration. And if you have any more queries, please notify us in a timely manner. 

Best wishes,

Mr. Zhong

---

## [Editor Report · Decision Letter 1]

18 Aug 2020

LncRNA MIR4435-2HG mediates cisplatin resistance in HCT116 cells by regulating Nrf2 and HO-1

PONE-D-19-25514R1

Dear Dr. Zhong,

We’re pleased to inform you that your manuscript has been judged scientifically suitable for publication and will be formally accepted for publication once it meets all outstanding technical requirements.

Kind regards,

Salvatore V Pizzo

Academic Editor

PLOS ONE
---

## [Editor Report · Acceptance letter]

12 Nov 2020

PONE-D-19-25514R1 

LncRNA MIR4435-2HG mediates cisplatin resistance in HCT116 cells by regulating Nrf2 and HO-1 

Dear Dr. Zhong:

I'm pleased to inform you that your manuscript has been deemed suitable for publication in PLOS ONE. Congratulations! Your manuscript is now with our production department. 

Kind regards, 

on behalf of

Dr. Salvatore V Pizzo 

Academic Editor

PLOS ONE